# RANS Modeling of Turbulent Flow and Heat Transfer in a Droplet-Laden Mist Flow through a Ribbed Duct

**Maksim A. Pakhomov *** and **Viktor I. Terekhov**

Laboratory of Thermal and Gas Dynamics, Kutateladze Institute of Thermophysics, Siberian Branch of Russian Academy of Sciences, Acad. Lavrent'ev Avenue, 1, 630090 Novosibirsk, Russia
* Correspondence: pakhomov@ngs.ru

**Abstract:** The local structure, turbulence, and heat transfer in a flat ribbed duct during the evaporation of water droplets in a gas flow were studied numerically using the Eulerian approach. The structure of a turbulent two-phase flow underwent significant changes in comparison with a two-phase flow in a flat duct without ribs. The maximum value of gas-phase turbulence was obtained in the region of the downstream rib, and it was almost twice as high as the value of the kinetic energy of the turbulence between the ribs. Finely dispersed droplets with small Stokes numbers penetrated well into the region of flow separation and were observed over the duct cross section; they could leave the region between the ribs due to their low inertia. Large inertial droplets with large Stokes numbers were present only in the mixing layer and the flow core, and they accumulated close to the duct ribbed wall in the flow towards the downstream rib. An addition of evaporating water droplets caused a significant enhancement in the heat transfer (up to 2.5 times) in comparison with a single-phase flow in a ribbed channel.

**Keywords:** droplet-laden flow; ribbed duct; heat transfer; evaporation; RANS; RSM





## 1. Introduction

The intensification of heat transfer in the internal cooling channels of gas turbine (GT) blades remains one of the key problems due to the constant growth in the inlet gas temperature of the GT. This temperature already reaches 2000 K and significantly exceeds the allowable temperatures for the long-term operation of the blades and power equipment of gas turbines [1–4]. Therefore, cooling the working surfaces of heat-loaded elements is an important and urgent problem of heat transfer. Various cooling methods (film cooling, jet impingement cooling, internal convective cooling, thermal barrier coatings, and spray cooling by the evaporation of various atomized droplets) have been developed for the effective thermal protection of working surfaces and increasing the operating times of power equipment elements. Internal convective cooling is a reliable and simple method for efficient cooling and heat removal from the GT heat-loaded elements.

One of the most effective methods for increasing heat transfer is the use of passive heat transfer intensifiers with various surface shapes. The use of various ribs or obstacles installed on a duct wall is one of the most effective ways to increase heat transfer (see monographs [5–7]). The rib height, $h$; duct height, $H$; rib pitch, $p$; obstacle shape; rib-to-channel height expansion ratio, $ER = h/H$; pitch-to-height ratio, $p/h$; and some other factors have a great effect on the formation and development of the recirculation region and heat transfer in such flows.

The heat transfer enhancement (HTE) of ribbed ducts (by 2–5 times) is accompanied by a significant increase in the pressure drop (of more than ten times) for most of these surfaces [1,3,4]. Two-dimensional obstacles most often have the form of ribs and protrusions of various configurations located at different angles to the flow on the duct walls [4–7]. They deflect and mix the flow, give rise to multiscale separated flows, and generate vorticity

and three-dimensional velocity gradients [4–7]. A two-phase flow around two-dimensional obstacles is one of the most common cases of shear separated flow for flows with both solid particles [8–12] and gas bubbles [13,14]. We should note that all of the abovementioned works were performed without taking into account the interfacial heat transfer between a dispersed phase (solid particles, droplets, and gas bubbles) and carrier fluid flow or in gas–liquid flows. The state-of-the-art research studying the movements and interactions of two-phase gas-dispersed flows with various obstacles was reviewed in [12,15].

The use of the latent heat of phase transition during the evaporation of droplets leads to a significant increase in HTE (up to several times) in comparison with conventional single-phase forced convection. Studies of the flow structure, friction, and heat transfer in a flow around ribs of various shapes with two-phase mist/steam and gas-droplet flows were carried out in several experimental [16–18] and numerical works [17,19–21].

The heat transfer in the case of a gas-droplet flow between two ribs was studied experimentally in [17]. The study was carried out with an initial mass fraction of water droplets of $M_{L1}$ = 15%. Their initial diameter was $d_1$ = 50–60 µm, the flow Reynolds number based on the mean mass flow velocity at the inlet and the hydraulic diameter was Re = $U_m D_h / \nu$ = (0.8–2.4) × $10^4$, the heat flux density was $q_W$ = const = 2.6 kW/m$^2$, and $p/h$ = 10 and 20 for the ribs installed at an angle to the free-stream flow of $\varphi$ = 90°. Heat transfer measurements in the case of a gas-droplet flow between two ribs in a system of continuous V-shaped ribs and broken V-shaped ribs were carried out in [18]. The study was carried out at $M_{L1}$ = 10%, $d_1$ = 50–60 µm, Re = (0.8–2.4) × $10^4$, $q_W$ = const = 1–10 kW/m$^2$, $p/h$ = 10 and 20, and $\varphi$ = 45°.

Numerical simulations [17,19–21] were performed using the commercial CFD package ANSYS Fluent using isotropic $k$–$\varepsilon$ (in [17,19]) and $k$–$\omega$ SST (in [20,21]) turbulence models. The effect of ribs installed at an angle ($\varphi$ = 45°) to the flow on the heat transfer in a two-phase flow was studied numerically in [22–24]. The predictions [17,20] were carried out for a two-phase flow in a smooth duct, and in [19,21] they were made in a U-shaped duct. Computations [19] were carried out for the following range of initial parameters: $M_{L1}$ = 2%, $d_1$ = 5 µm, Re = (0.5–4) × $10^4$, $q_W$ = const = 10 kW/m$^2$, and $p/h$ = 10 in a gas-droplet flow. The range of variation in the initial parameters in [20] was as follows: $M_{L1}$ = 1%, $d_1$ = 10 µm, Re = (1–6) × $10^4$, $q_W$ = const = 4.8 kW/m$^2$, and $p/h$ = 10 in the flow of steam water droplets.

An analysis of previous works allowed us to draw the following conclusions. Note that the studies in [16,21] were performed for a single-component mist/steam flow. In this area, the first steps have been taken, which revealed the great potential of such a cooling method. There are an extremely limited number of papers in the literature concerned with the study of heat transfer in a turbulent droplet-laden flow in a ribbed duct. In these works, a significant HTE up to three times was experimentally and numerically shown in comparison with a single-phase flow in a smooth duct with a fixed Reynolds number in the flow. All numerical works [17,19–21] used the RANS approach and isotropic turbulent models (ITM). The use of such models for the simulation of a complicated vortex turbulent flow has a number of limitations [22–24], even for a simpler case of flow in a backward-facing step. The Euler–Lagrangian approach was used to model the dynamics and heat transfer in the two-phase flow. Research in this direction should be deepened and detailed. In addition to the flow and turbulent characteristics, heat transfer should also be studied.

In the present study, the authors used the Euler–Euler approach for flow and heat transfer simulation in the dispersed phase [25]. The turbulent characteristics of the carrier phase were predicted using the elliptical blending Reynolds stress model (RSM). This approach allows the partial elimination of the problems associated with the significant anisotropy of turbulent velocity fluctuations for the flows with recirculating regions [22–24]. This work is aimed at the numerical study of the effect of droplet evaporation on the flow, turbulence, and heat transfer in a ribbed duct in comparison with a smooth one.

## 2. Mathematical Models

The paper considers the flow dynamics and heat transfer in 2D two-phase gas-droplet turbulent flow in the presence of interfacial heat transfer between the ribs. The two-fluid Euler approach is used to describe the flow dynamics and heat and mass transfer in the gaseous and dispersed phases [26,27]. The carrier phase turbulence is predicted using the elliptical Reynolds stress model [28], taking into account the effect of droplets [29,30]. The dispersed phase (water droplets) is described using steady-state continuity equations, two momentum equations, and energy equations. The authors used their own in-house code for all numerical simulations presented in this paper.

### 2.1. Governing Equations for the Two-Phase Turbulent Mist Phase

The set of incompressible steady-state 2D RANS equations of the carrier phase includes continuity equations, two momentum equations (in streamwise and transverse directions), energy equations, and steam diffusion into the binary air–steam medium [25]. The effect of evaporating water droplets on the motion and heat transfer in the carrier phase (air) is considered using the sink or source terms.

$$
\begin{aligned}
&\rho \frac{\partial U_j}{\partial x_j} = \frac{6J}{d}\Phi \\
&\frac{\partial (U_i U_j)}{\partial x_i} = -\frac{\partial P}{\rho \partial x_i} + \frac{\partial}{\partial x_j}\left(\nu \frac{\partial U_i}{\partial x_j} - \left\langle u_i' u_j' \right\rangle\right) - (U_i - U_{Li})\frac{M_L}{\tau} \\
&\frac{\partial (U_i T)}{\partial x_i} = \frac{\partial}{\partial x_i}\left(\frac{\nu}{\mathrm{Pr}}\frac{\partial T}{\partial x_i} - \left\langle u_j' t \right\rangle\right) + D_T \frac{(C_{PV} - C_{PA})}{C_P}\left(\frac{\partial K_V}{\partial x_i}\frac{\partial T}{\partial x_i}\right) - \frac{6\Phi}{\rho C_P d}\left[\alpha(T - T_L) + JL\right] \\
&\frac{\partial (U_i K_V)}{\partial x_i} = \frac{\partial}{\partial x_i}\left(\frac{\nu}{\mathrm{Sc}}\frac{\partial K_V}{\partial x_i} - \left\langle u_j' k_V \right\rangle\right) + \frac{6J\Phi}{d} \\
&\rho = P/(R_g T)
\end{aligned}
\tag{1}
$$

Here, $U_i$ ($U_x \equiv U$, $U_y \equiv V$) and $u_i'$ ($u_x' \equiv u'$, $u_y' \equiv v'$) are components of mean gas velocities and their pulsations; $x_i$ are projections on the coordinate axis; $2k = \langle u_i u_i \rangle = u'^2 + v'^2 + w'^2 \approx u'^2 + v'^2 + 0.5(u'^2 + v'^2) \approx 1.5(u'^2 + v'^2)$ is the kinetic energy of gas-phase turbulence; $\tau = \rho_L d^2/(18\rho\nu W)$; $W = 1 + \mathrm{Re}_L^{2/3}/6$ is the particle relaxation time, taking into account the deviation from the Stokes power law; and $\mathrm{Re}_L = |\mathbf{U} - \mathbf{U}_L|d/\nu$ is the Reynolds number of the dispersed phase.

The turbulent heat $\left\langle u_j' t \right\rangle = -\frac{\nu_T}{\mathrm{Pr}_T}\frac{\partial T}{\partial x_j}$, and the mass $\left\langle u_j' k_V \right\rangle = -\frac{\nu_T}{\mathrm{Sc}_T}\frac{\partial K_V}{\partial x_j}$ fluxes in the gas phase are predicted using simple eddy diffusivity (Boussinesq hypothesis). The constant value of the turbulent Prandtl and Schmidt numbers, $\mathrm{Pr}_T$ and $\mathrm{Sc}_T$ equal to 0.9, is used in this work.

### 2.2. Evaporation Model

The set of Eulerian Equation (1) of two-phase flow is supplemented by the equations of heat transfer on the droplet surface and the conservation equation of steam on the surface of the evaporating droplet [31]. It is assumed that the temperature over the droplet radius is constant [31].

$$
\lambda_L \left(\frac{\partial T_L}{\partial y}\right)_L = \alpha(T - T_L) - JL, \quad \alpha = \frac{\alpha_P}{1 + C_P(T - T_L)/L} = \frac{\alpha_P}{1 + \mathrm{Ja}}
\tag{2}
$$

$$
J = JK_V^* - \rho D \left(\frac{\partial K_V}{\partial y}\right)_L
\tag{3}
$$

Here, $\lambda_L$ is the coefficient of heat conductivity of the droplet; $\alpha$ and $\alpha_P$ are the heat transfer coefficient for the evaporating droplet and non-evaporating particle, respectively; $T_L$ is the temperature of the droplet; $J$ is the mass flux of steam from the surface of the evaporating droplet; $L$ is the latent heat of evaporation; $\rho$ is the density of the gas–steam mixture; $D$ is the diffusion coefficient; and $K_V^*$ is the steam mass fraction at the "steam-gas mixture–droplet" interface, corresponding to the saturation parameters at droplet temperature $T_L$. Subscript "$L$" corresponds to the parameter on the droplet surface. The

Jacob number, $Ja = C_P(T - T_L)/L$, is the ratio of sensible heat to latent heat during droplet evaporation. It characterizes the rate of the evaporation process and is the reciprocal of the Kutateladze number, Ku. For our conditions, the Jakob number is $Ja \leq 0.01$.

The expression for the diffusional Stanton number has the form

$$\text{St}_D = -\rho D \left( \frac{\partial K_V}{\partial y} \right)_L / [\rho \mathbf{U}(K_V^* - K_V)] \tag{4}$$

We can insert Equation (4) into Equation (3). Equation (3) can be written in the form

$$J = \text{St}_D \rho \mathbf{U} b_{1D}, \tag{5}$$

where $b_{1D} = \left( K_V^* - K_V \right)/\left( 1 - K_V^* \right)$ is the diffusion parameter of vapor (steam) blowing, determined with the use of a saturation curve.

A droplet evaporates at the saturation temperature, and the temperature distribution inside a droplet is uniform. The droplet temperature along the droplet radius remains constant because the Biot number is $Bi = \alpha_L d_1/\lambda_L << 1$ and the Fourier number is $Fo = \tau_{eq}/\tau_{evap} << 1$. Here, $\tau_{eq}$ is the period when an internal temperature gradient inside a droplet exists, and $\tau_{evap}$ is the droplet's lifetime. In this case, a droplet evaporates at the saturation temperature, and the temperature distribution inside a droplet is uniform.

### 2.3. The Elliptic Blending Reynolds Stress Model (RSM) for the Gas Phase

In the present study, the low-Reynolds-number elliptic blending RSM of [28] is employed. The transport equations for $\left\langle u_i' u_j' \right\rangle$ and the kinetic energy dissipation rate, $\varepsilon$, are written in the following general form:

$$\frac{\partial \left( U_j \left\langle u_i' u_j' \right\rangle \right)}{\partial x_j} = P_{ij} + \phi_{ij} - \varepsilon_{ij} + \frac{\partial}{\partial x_l} \left( \nu \delta_{lm} + \frac{C_\mu T_T}{\sigma_k} \left\langle u_l' u_m' \right\rangle \right) \frac{\partial \left\langle u_i' u_j' \right\rangle}{\partial x_m} - A_L, \tag{6}$$

$$\frac{\partial \left( U_j \varepsilon \right)}{\partial x_j} = \frac{1}{T_T} \left( C_{\varepsilon 1}' P_2 - C_{\varepsilon 2} \varepsilon \right) + \frac{\partial}{\partial x_l} \left( \nu \delta_{lm} + \frac{C_\mu T_T}{\sigma_\varepsilon} \frac{\partial \varepsilon}{\partial x_m} \right) - \varepsilon_L, \tag{7}$$

$$\beta - L_T^2 \nabla^2 \beta = 1, \quad \phi_{ij} = \left( 1 - \beta^2 \right) \phi_{ij}^W + \beta^2 \phi_{ij}^H \tag{8}$$

Here, $P_{ij}$ is the stress-production term, $T_T$ and $L_T$ are the turbulent time and geometrical macroscales, and $\phi_{ij}$ is the velocity–pressure–gradient correlation, well-known as the pressure term. The blending model (8) presented in [28] is used to predict $\phi_{ij}$ in Equations (6) and (7), where $\beta$ is the blending coefficient, which goes from zero at the wall to unity far from the wall; $\phi_{ij}^H$ is the "homogeneous" part (valid away from the wall) of the model; and $\phi_{ij}^W$ is the "inhomogeneous" part (valid in the wall region).

The other constants and functions of the turbulence model are presented in detail in [28]. The last terms of the system of Equations (6) and (7), $A_L$ and $\varepsilon_L$, represent the effects of particles on carrier phase turbulence [29,30].

### 2.4. Governing Equations for the Dispersed Phase

The set of incompressible steady-state 2D governing mean equations for the dispersed phase consists of continuity equation, two momentum equations (in streamwise and transverse directions), energy equations.

$$\frac{\partial \left( \rho_L \Phi U_{Lj} \right)}{\partial x_j} = -\frac{6J\Phi}{d},$$

$$\frac{\partial \left( \rho_L \Phi U_{Lj} U_{Li} \right)}{\partial x_j} + \frac{\partial \left( \rho_L \Phi \left\langle u_{Li} u_{Lj} \right\rangle \right)}{\partial x_j} = \Phi (U_i - U_{Li}) \frac{\rho_L}{\tau} + \Phi \rho_L g - \frac{1}{\tau} \frac{\partial \left( \rho_L D_{Lij} \Phi \right)}{\partial x_j} - \frac{\partial (\Phi P)}{\partial x_i}, \tag{9}$$

$$\frac{\partial(\rho_L \Phi U_{Lj} T_{Li})}{\partial x_j} + \frac{\partial}{\partial x_j}\left(\rho_L \Phi \langle \theta u_{Lj} \rangle\right) = \Phi(T_i - T_{Li})\frac{\rho_L}{\tau_\Theta} - \frac{1}{\tau_\Theta}\frac{\partial\left(\rho_L D^{\Theta}_{Lij} \Phi\right)}{\partial x_j}. \tag{10}$$

where $D_{Lij}$ and $D^{\Theta}{}_{Lij}$ are the turbulent diffusivity tensor and the particle turbulent heat transport tensor [29,30], $\tau_\Theta = C_{PL}\rho_L d^2/(12\lambda Y)$ is the thermal relaxation time, and $Y = \left(1 + 0.3\text{Re}_L^{1/2}\text{Pr}^{1/3}\right)$.

The set of governing mean equations for the dispersed phase (8–10) is completed by the kinetic stress equations, temperature fluctuations, and turbulent heat flux in the dispersed phase, which are in the form presented in [29,30].

The volume fraction of the dispersed phase is lower ($\Phi_1 < 10^{-4}$), and the droplets are finely dispersed ($d_1 < 100$ µm); therefore, the effects of interparticle collisions and break-up are neglected [25,32,33]. Droplet bag break-up is observed at We $= \rho(\mathbf{U}_S - \mathbf{U}_L)^2 d/\sigma \geq$ We$_{cr}$ $= 7$ [33]. Here, $U_S = U + \langle u'_S \rangle$ and $\mathbf{U}_L$ are the gas velocity seen by the droplet [34] and the mean droplet velocity, respectively, $U$ is the mean gas velocity (derived directly from the RANS predictions), $\langle u'_S \rangle$ is the drift velocity between the fluid and the particles [34], and $\rho$ and $\rho_L$ are the densities of the gas and dispersed phases. For all droplet sizes investigated in the present paper, the Weber number is very small (We << 1). Droplet fragmentation at its contact with a duct wall also is not considered. The effect of break-up and coalescence in the two-phase mist flow can be neglected due to a low droplet volume fraction at the inlet ($\Phi_1 = M_{L1}\rho/\rho_L < 2 \times 10^{-4}$). Here, $M_{L1}$ is the initial droplet mass fraction, and $\rho_L$ is the density of the dispersed phase.

A scheme of the flow is shown in Figure 1. A similar Euler approach was used by the authors to describe gas-droplet axisymmetric flows behind a sudden pipe expansion [25] and behind a backward-facing step in a flat duct [35].

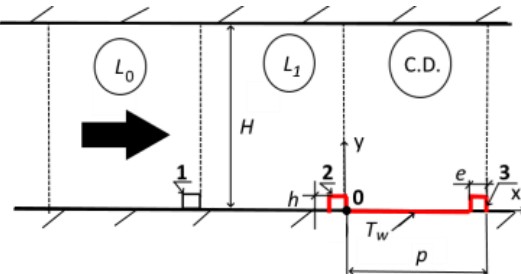

**Figure 1.** Scheme of flow in a two-phase turbulent flow in a ribbed flat duct (not to scale). Abbreviations $L_0$ and $L_1$ are the computational domains for the preliminary simulations; C.D. is the computational domain; and 1, 2, and 3 are the 1st, 2nd, and 3rd ribs.

## 3. Numerical Solution and Model Validation

### 3.1. Numerical Solution

The solution was obtained using the finite volume method on staggered grids. The QUICK procedure of the third order of accuracy was used for the convective terms. Central differences of the second order of accuracy were used for diffusion fluxes. The pressure field was corrected according to the agreed finite volume SIMPLEC procedure. The components of the Reynolds stress of the carrier fluid phase were simulated according to the method proposed in [36]. The components of the Reynolds stress were determined at the same points along the control volume faces as the corresponding components of the average velocity of the carrier phase. The computational grid consisted of rectangular cells. It was inhomogeneous and thickened towards all solid walls, which was necessary to resolve the details of the turbulent flow in the near-wall zone (see Figure 2). In the viscous sublayer, at least 10 computational volumes (CVs) were set. The correct simulation of sharp gradients of two-phase flow parameters was necessary. The coordinate transformation given in [37] was suitable for such a two-dimensional boundary layer problem.

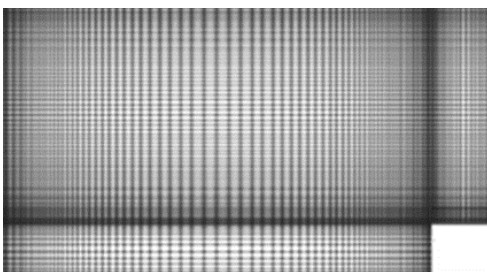

**Figure 2.** Computational mesh "medium" (not to scale).

All predictions were carried out on a "medium" grid containing $256 \times 120$ control volumes (CVs). The first computational cell was located at a distance from the wall of $y_+ = u_* y / \nu \approx 0.5$ (the friction velocity $u_*$ was determined for a single-phase air flow with other identical parameters). Additionally, simulations were carried out on grids containing "coarse" $128 \times 60$ and "fine" $512 \times 200$ CVs. The difference in the results of the calculations of the wall friction coefficient (a) and the Nusselt number (b) for two-phase flow did not exceed 0.1% (see Figure 3). The Nusselt number at $T_W = $ const was determined by the formula:

$$\mathrm{Nu} = -(\partial T / \partial y)_W H / (T_W - T_m),$$

where $T_W$ and $T_m$ are the wall and the mass-averaged temperatures of the gas in the corresponding cross section.

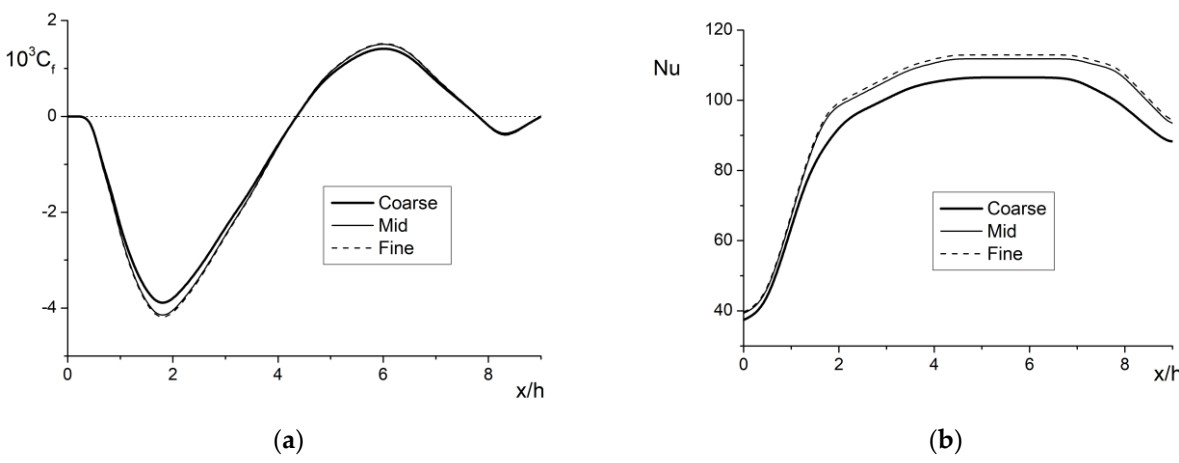

(a)                                                         (b)

**Figure 3.** Grid independence test for the wall friction (**a**) and heat transfer (**b**) distributions along the duct length. $M_{L1} = 5\%$, $d_1 = 15$ μm, Re $= HU_{m1}/\nu = 1.6 \times 10^4$.

Periodic boundary conditions were set at the inlet of the computational domain. Initially, a single-phase fully hydrodynamically developed air flow was supplied to the inlet to the computational domain $L_0 = 10p$, where $p$ is the rib pitch (the spacing between upstream and downstream ribs). The 1st rib was installed at the end of this domain. The output parameters from section $L_0$ were the input values for section $L_1 = 10p$, located between the 1st and 2nd ribs (see Figure 1). All simulations were performed for the two-dimensional case of a gas-droplet flow for the 2nd and 3rd obstacles. Drops were fed into a single-phase turbulent air flow along the entire cross section of the duct in the inlet cross section behind the 2nd rib. The initial temperatures of the gas and dispersed phases at the inlet to the computational domain were $T_1 = T_{L1} = 293$ K. The boundary condition $T_W = $ const $ = 373$ K was set on the ribbed wall; the opposite smooth (without obstacles) wall of the flat duct was adiabatic. The entire ribbed duct surface and all the ribs were heated to eliminate the influence of the possible formation of liquid spots during the deposition of droplets on the wall from a two-phase mist flow. The impermeability and no-slip conditions

for the gas phase were imposed on the duct walls. For the dispersed phase on the duct wall, the boundary condition of the "absorbing wall" [30] was used when a droplet did not return to the flow after contact with the wall surface. All droplets deposited from two-phase flow onto the wall momentarily evaporated. Thus, the pipe surface was always dry, and there was no liquid film or spots of deposited droplets formed on the wall [25,31,35]. This assumption for the heated surface is valid (see, for example, papers [25,35]). Furthermore, this condition is valid if the temperature difference between the wall and the droplet is greater than $T_W - T_L \geq 40\ K$ [38]. In the outlet cross section, the conditions for the equality to zero of the derivatives of all variables in the streamwise direction were set.

*3.2. Model Validation*

At the first stage, a comparison with the data of recent LDA measurements [39] for a single-phase air flow in the presence of ribs was performed. The results of the experiments and predictions are shown in Figure 4. This figure shows comparisons of measured and predicted data in the form of transverse profiles of mean longitudinal velocity, $U/U_{m1}$ (a), and the velocity of its fluctuations, $u'/U_{m1}$ (b), along the duct length. The averaged and fluctuating components of the streamwise velocity were normalized by the value of the average mass velocity of a single-phase flow at the duct inlet $U_{m1}$. Comparisons with the data of [39] were made for the 17th and 18th obstacles. The height of the duct with a square cross section was $H$ = 60 mm. The profiles of the mean longitudinal velocity component agreed well with the experimental data (the difference did not exceed 5–7%). The agreement between the measurements and numerical predictions for longitudinal velocity pulsations was also quite good (the difference did not exceed 10%) except for the near-wall region.

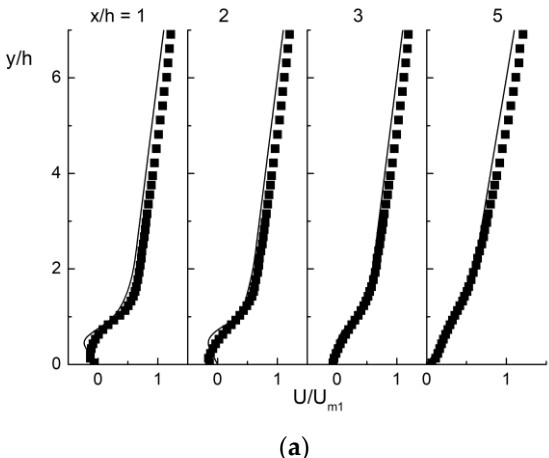
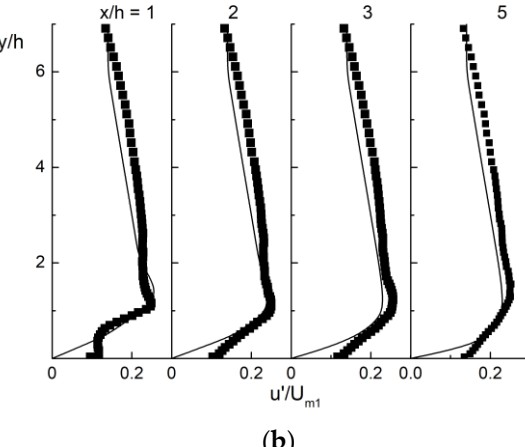

(**a**)                                                                                                                    (**b**)

**Figure 4.** Profiles of the mean longitudinal velocity (**a**) and its fluctuations (**b**) in the flow around a two-dimensional square obstacle. (**a**): $h/H$ = 0.067, Re = $U_{m1}H/\nu$ = 5 × 10⁴, $p/h$ = 9, $H$ = 60 mm, $h = e$ = 4 mm, $p$ = 36 mm, $U_{m1}$ = 12.5 m/s. The symbols are the measurements of [39]; the lines are the authors' simulations.

The results of measurements [40] and RANS numerical simulations with various isotropic turbulence models ($k$–$\varepsilon$, $v2f$, and $k$–$\omega$ shear stress tensor (SST)) [41] for the flow in the ribbed duct were used for heat transfer comparisons. Satisfactory agreement with the data of other authors for a single-phase flow around a two-dimensional obstacle was obtained (the maximum differences did not exceed 15%), except for the duct cross section near the upstream obstacle at $x/h < 2$ (see Figure 5). Here, Nu is the Nusselt number in a ribbed duct and $Nu_0$ is the Nusselt number in a smooth duct for a single-phase flow. The Nusselt number at a constant value of heat flux density ($q_W$ = const) is determined by the formula:

$$Nu = q_W H / [\lambda (T_W - T_m)].$$

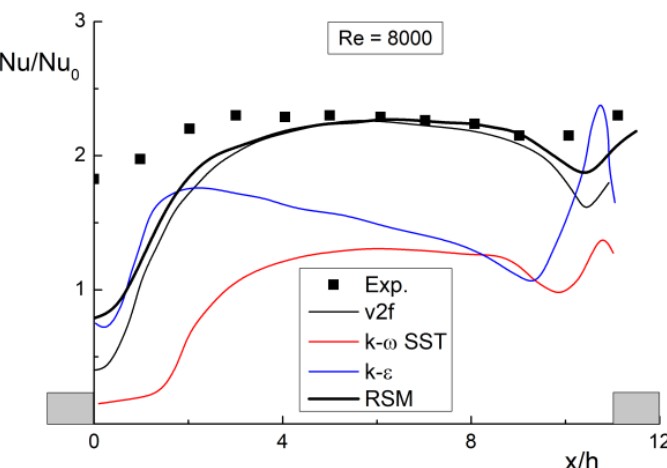

**Figure 5.** Distribution of heat transfer enhancement ratio in the flow around a two-dimensional square obstacle. $Re_D = 0.8 \times 10^4$, $h = e = 4$ mm, $p = 40$ mm, $H = 30$ mm, $H/h = 7.5$, $T_1 = 300$ K, $q_W = 1$ kW/m$^2$, Tu$_1$ = 5%. The symbols are the experiments of [40]; the lines are predictions: *v2f*, *k–ω* SST, and *k–ε* are predictions of [41], and RSM is the authors' simulations. Reprinted with permission from (Liu, J. et al.).

Comparisons with the data [40,41] were made for the 7th and 8th obstacles. All predictions were carried out for a flat duct with a square cross section and a height of $H = 30$ mm.

## 4. The RANS Results and Discussion

All 2D numerical simulations were carried out for a mixture of air with water drops at the duct inlet for the case of a downward two-phase flow at atmospheric pressure. Ribs were installed on the "bottom" wall of the flat duct. All simulations were performed for the flow around the system of the 2nd and 3rd obstacles. The computational domain included two square ribs with a height of $h = 4$ mm. The height of a smooth duct was $H = 40$ mm ($H/h = 10$), and the distance between two ribs was $p/h = 5$–12. The mass-average gas velocity in the inlet cross section in the computational domain varied within $U_{m1} = 5$–20 m/s, and the Reynolds number for the gas phase, constructed from the mass-average gas velocity at the inlet and the duct height, was $Re_H = HU_{m1}/\nu \approx (0.6$–5$) \times 10^4$. The initial average droplet diameter was $d_1 = 5$–50 μm, and their mass concentration was $M_{L1} = 0$–10%. The initial temperature of the gaseous and dispersed phases was $T_1 = T_{L1} = 293$ K.

A turbulent flow is 3D in nature. Nevertheless, there are many cases when it is possible to use a 2D approach to describe a quasi-two-dimensional turbulent flow, for example, if the duct width, $Z$, is much greater than its height, $H$ ($Z/H > 10$). The authors of [42] recommended the consideration of the turbulent solid particle-laden flow in a backward-facing step in a flat channel as two-dimensional due to the large aspect ratio of $Z/H$.

### 4.1. Flow Structure

The streamlines for a gas-droplet flow around the system of two ribs are shown in Figure 6. The complex vortex structures of the averaged flow between two ribs are clearly visible. The formation of two regions of the flow recirculation is shown. The first large recirculation region formed behind the upstream rib due to the separation of the two-phase flow at the backward-facing step (BFS). A small corner vortex was located at the end of the reverse step. The second one formed due to the droplet-laden flow separation before the downstream rib (forward-facing step (FFS)) when the fluid flow left the cell between the two ribs. It was much shorter than the previous one.

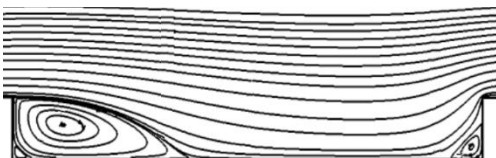

**Figure 6.** The streamlines for a gas-droplet flow between two ribs. Re $= U_{m1}H/\nu = 1.6 \times 10^4$, $h/H = 0.1$, $p/h = 10$, $H = 40$ mm, $h = 4$ mm, $p = 40$ mm, $U_{m1} = 6$ m/s, $T_1 = 293$ K, $T_W = 373$ K, $d_1 = 15$ µm, $M_{L1} = 0.05$.

Figure 7 shows the profiles of the average longitudinal velocities, $U/U_{m1}$ (a); turbulent kinetic energy (TKE), $k/U_{m1}^2$ (b); and gas-phase temperature, $\Theta = (T_W - T)/(T_W - T_{1,m})$, in a single-phase flow ($M_{L1} = 0$), a gas-droplet flow at $M_{L1} = 0.05$, and liquid drops $\Theta_L = (T_{L,max} - T_L)/(T_{L,max} - T_{L1})$ (c) as well as vorticity, $\Omega_z = \omega_z h/U_{m1}$ (d). The solid lines are the single-phase flow at $M_{L1} = 0$, the dotted line is the gas phase at $M_{L1} = 0.05$, and the dash-dot line is the dispersed phase behind the two-dimensional obstacle. Here, $T_{L,max}$ и $T_{L1,m}$ are the droplet temperatures, which were highest in the corresponding cross section and at the inlet.

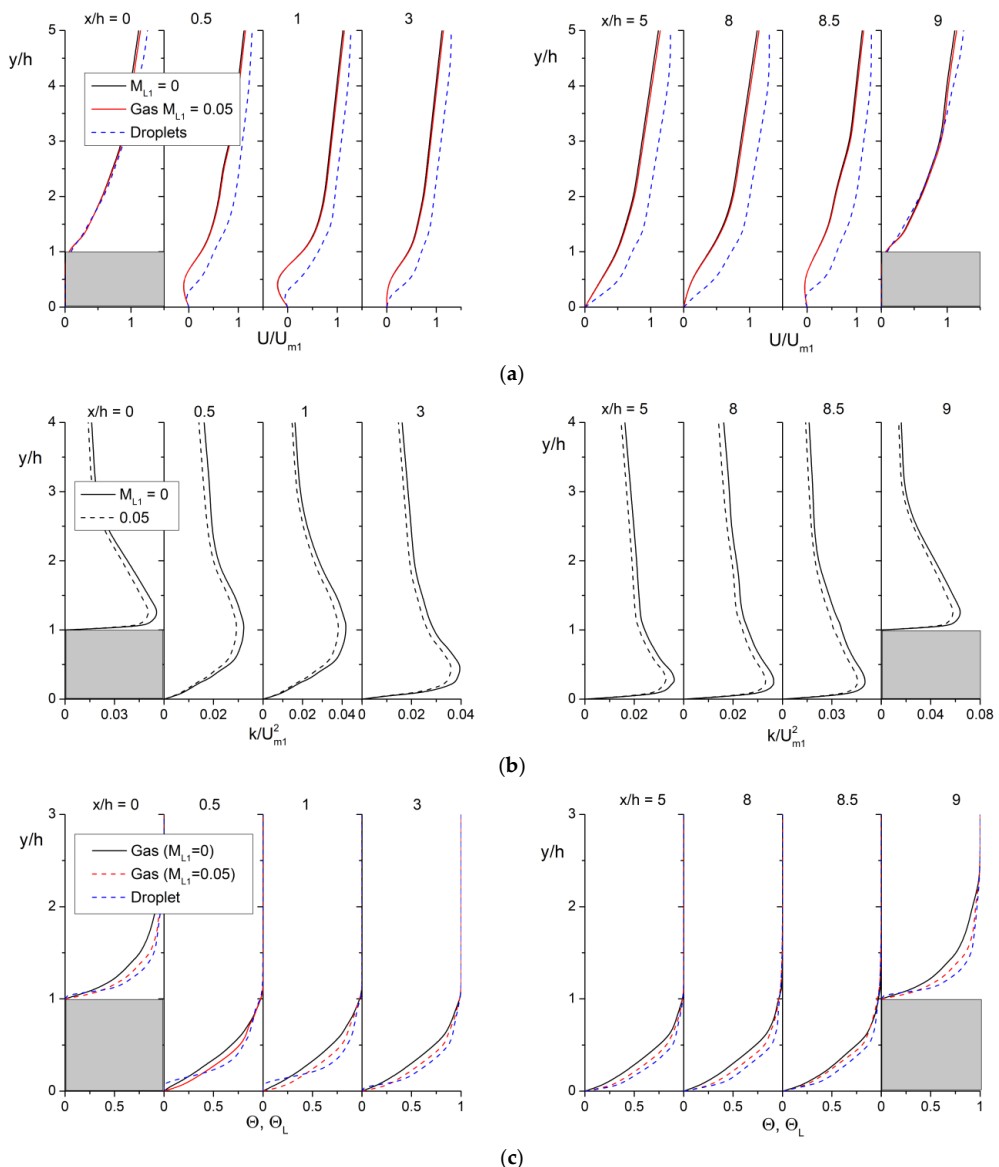

**Figure 7.** *Cont.*

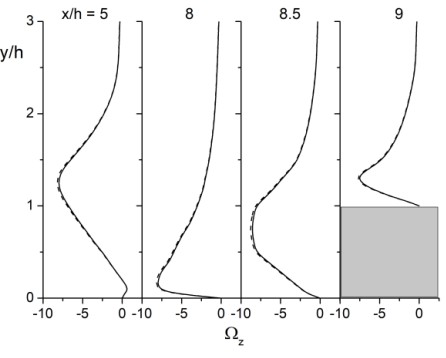 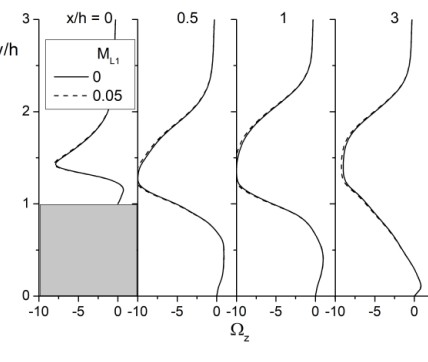

**(d)**

**Figure 7.** Transverse profiles of averaged streamwise velocities (**a**), turbulent kinetic energy (**b**), and gas-phase temperature in a single-phase flow ($M_{L1}$ = 0), a gas-droplet flow at $M_{L1}$ = 0.05, and liquid drops (**c**) as well as vorticity (**d**). Re = $1.6 \times 10^4$, $h/H$ = 0.1, $p/h$ = 10, $d_1$ = 15 μm, $M_{L1}$ = 0.05.

The structure of a turbulent two-phase flow showed significant changes when flowing around a system of obstacles installed on one of the duct walls. The profiles of the averaged streamwise velocity components of the gaseous and dispersed phases were similar to those for the single-phase flow regime (see Figure 7a). The gas velocity in the gas-droplet flow was slightly (≤3%) ahead of the single-phase flow velocity. The drop velocity had the greatest value for the downward flow due to their inertia. Two regions with negative values for the longitudinal velocity of the gas-droplet flow are shown, which were confirmed by the data in Figure 6. The length of the main recirculation zone of the flow was $x_{R1} \approx 4.1h$, and the length of the second recirculation region in front of the step ahead was $x_{R2} \approx 1.1h$. The lengths of the recirculation zones were determined from the zero value of the flow velocity.

Figure 7b shows the transverse distributions of the kinetic energy (TKE) of carrier phase turbulence for a 2D flow. The TKE was calculated by the formula for a two-dimensional flow:

$$2k = \langle u_i' u_i' \rangle = u'^2 + v'^2 + w'^2 \approx u'^2 + v'^2 + 0.5\left(u'^2 + v'^2\right) \approx 1.5\left(u'^2 + v'^2\right)$$

The highest turbulence values were obtained for the mixing layer. The level of kinetic energy of turbulence increased as the downstream obstacle approached. The maximum value of gas-phase turbulence was obtained at $x/h$ = 9 (the upper corner of the downstream rib), and it was almost twice as high as the values for the TKE between the ribs. The turbulence of the flow was associated with the flow around the obstacle.

The dimensionless temperature distributions of the single-phase flow and the gas and dispersed phases are shown in Figure 7c. All profiles in Figure 7c are qualitatively similar to each other. The gas temperature in the gas-droplet flow was lower than the corresponding value for a single-phase flow due to droplet evaporation. Let us note that the droplet temperature profile for the first two sections, $x/h$ = and 3, did not start from the wall ($y/h$ = 0) as for the gas phase, but it is shifted from the wall by a small distance towards the flow core. This is explained by the absence of droplets in the near-wall zone in the area of flow separation due to their evaporation close to the wall between the ribs. The non-dimensional vorticity profiles are given in Figure 7d. They were calculated using the well-known formula:

$$\omega_z = \frac{\partial V}{\partial x} - \frac{\partial U}{\partial y}. \tag{11}$$

The magnitudes of vorticity were mainly negative values (because $\frac{\partial V}{\partial x} \ll \frac{\partial U}{\partial y}$), except in the near-wall region inside the flow recirculation zone (see Figure 7d). The minimal values are shown in the outer shear layer of the separation zone and on the top wall of the downstream rib. The maximal positive value was observed close to the wall of the ribbed

wall. In the case of two-phase mist flow, the magnitude of vorticity was slightly higher than that of the single-phase flow (up to 4%).

Figure 8 shows the profiles of the dispersed-phase mass concentration, $M_L/M_{L1}$, for various droplet mass fractions (a) and their initial diameters (b). Obviously, due to the evaporation of droplets, their mass fraction decreased continuously, both streamwise and in traverse directions, when approaching the wall of the heated duct between the ribs. This was typical of the numerical data given in Figure 8a,b. The distributions of the mass fraction of droplets with changes in their initial amounts had qualitatively similar forms (see Figure 8a).

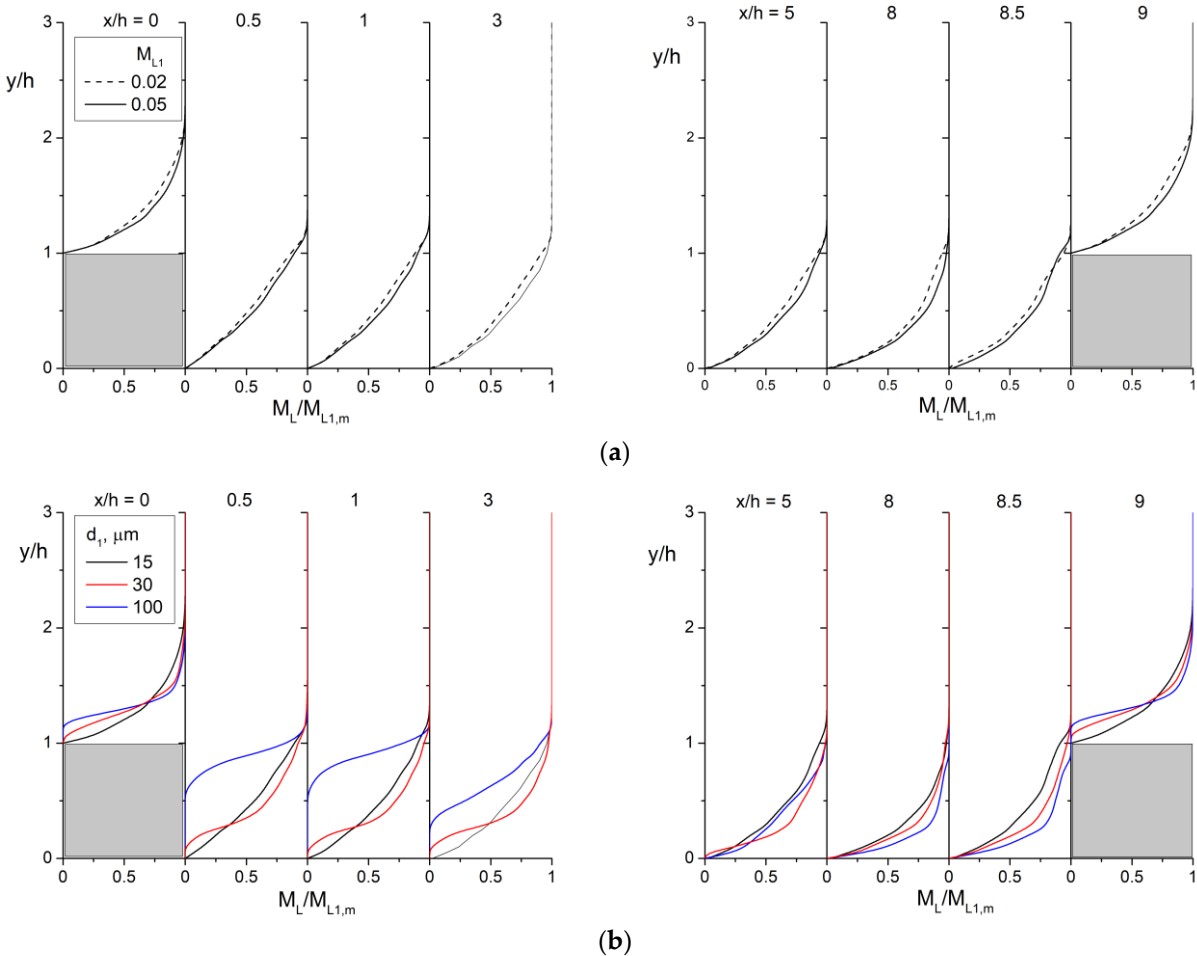

**Figure 8.** Transverse profiles of the water droplet mass fraction vs. the initial mass fraction (**a**) and the droplet diameter (**b**). Re = $1.6 \times 10^4$. (**a**) $d_1 = 15$ μm; (**b**) $M_{L1} = 0.05$.

A change in the initial diameter of the liquid droplets had a more complex effect on the course of the evaporation processes (see Figure 8b). In the flow core, this value trended toward the corresponding value at the inlet to the computational domain, and $M_L/M_{L1} \rightarrow 1$. This is explained by the almost complete absence of droplet evaporation. Fine particles at Stk < 1 penetrated into the region of flow separation and were observed over the entire cross section of the duct. Large inertial droplets ($d_1 = 100$ μm, Stk > 1) almost did not penetrate into the flow recirculation zone, and they were present in the mixing layer and the flow core. In the near-wall zone, large drops were observed only behind the reattachment point. The largest and inertial droplets ($d_1 = 100$ μm) accumulated in the near-wall region towards the downstream obstacle. Finely dispersed low-inertia droplets could leave the region between the two ribs due to their low inertia, while large drops

could not leave this region. This led to an increase in the droplet mass fraction in this flow region and towards the downstream obstacle.

In order to clearly display the flow structure in the inter-rib cavity, the contours of the nondimensional mean streamwise velocity, $U/U_{m1}$ (a), and the temperature, $\Theta = (T_W - T)/(T_W - T_{1,m})$ (b), in two-phase mist flow are shown in Figure 9. Large-scale and small-scale flow recirculation zones behind the upwind rib (BFS) and before the downstream rib (FFS) can be found in Figure 9a. The small corner vortex directly behind the upstream rib was observed. The length of the main recirculation zone of the flow was $x_{R1} \approx 4.1h$, and the length of the second recirculation region in front of the step ahead was $x_{R2} \approx 1.1h$. The lengths of the recirculation zones were determined from the zero value of the mean streamwise flow velocity ($U = 0$). In this region, the gas temperature increased, and it led to the suppression of heat transfer (see Figure 9b). These conclusions agree with the data of Figures 6 and 7a,c.

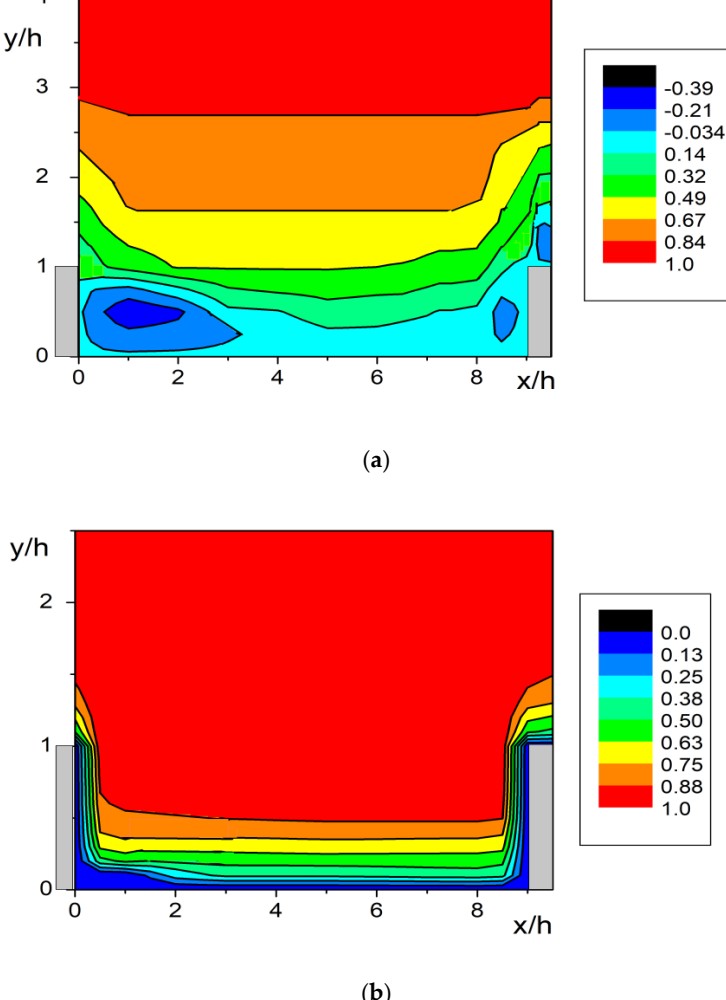

**Figure 9.** The contour plots of the mean streamwise velocity (**a**) and temperature (**b**). Re = $1.6 \times 10^4$, $d_1 = 15$ μm, $M_{L1} = 0.05$.

### 4.2. Heat Transfer

The influence of the initial mass fraction (a) and droplet diameter (b) of the dispersed phase on the Nusselt number distribution in a two-phase flow along the duct length is shown in Figure 10. A significant HTE in the two-phase mist flow (up to 2.5 times) compared to a single-phase flow in a ribbed channel was obtained with the addition of evaporating water drops into a single-phase gas flow (see Figure 10a). Droplets of the

minimum diameter ($d_1$ = 5 μm) evaporated most intensely, and the largest ones evaporated least intensely ($d_1$ = 100 μm) (see Figure 10b). The sizes of the zone of two-phase flow and the zone of HTE also decreased. This was an obvious fact for the evaporation of droplets in the two-phase mist flows, which was associated with a significant interface reduction; it was first shown by the authors of this work for a gas-droplet flow in a system of two-dimensional obstacles. Heat transfer was attenuated and trended toward the corresponding value for the single-phase flow in the region of flow separation for the most inertial droplets. These drops did not penetrate into the flow separation region behind the upstream rib (BFS). An increase in heat transfer was obtained in the region behind the point of flow reattachment. A decrease in heat transfer was shown in the section of flow separation towards the downstream rib (FFS). The most inertial droplets also did not leave the region between the two ribs and accumulated in front of the downstream obstacle.

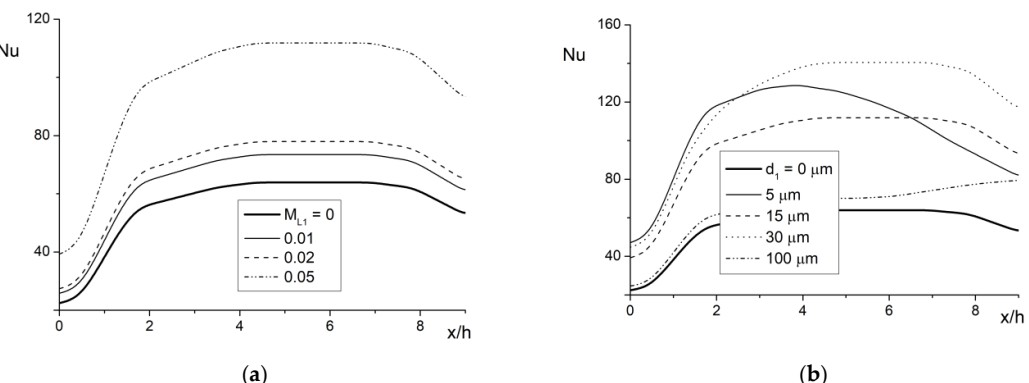

**Figure 10.** The effect on the heat transfer rate of the droplet mass fraction at the inlet (**a**) and their diameter (**b**). Re = $1.6 \times 10^4$. (**a**) $d_1$ = 15 μm; (**b**) $M_{L1}$ = 0.05.

The effect of the gas Reynolds number, Re, and the initial mass fraction of the dispersed phase, $M_{L1}$, on the thermal hydraulic performance parameter is shown in Figure 11. The wall friction coefficient, $C_f$, was calculated using the formula $C_f/2 = \tau_W / (\rho U_{m1}^2)$. Here, $Nu_0$ and $C_{f0}$ are the maximal Nusselt number and wall friction coefficient in the two-phase mist flow of a fully developed smooth duct, other conditions being equal. $Nu/Nu_0/(C_f/C_{f0})$ is the thermal hydraulic performance parameter. This is the ratio of the maximal Nusselt numbers divided by the maximal wall friction coefficient ratio. The ribbed surface provided a much better thermohydraulic performance than a smooth duct in the case of a droplet-laden turbulent mist flow, with other conditions being identical. This effect was quite pronounced at small Reynolds number values of Re < $10^4$. It should be noted that the wall friction coefficient ratio, $C_f/C_{f0}$, was taken to the first power.

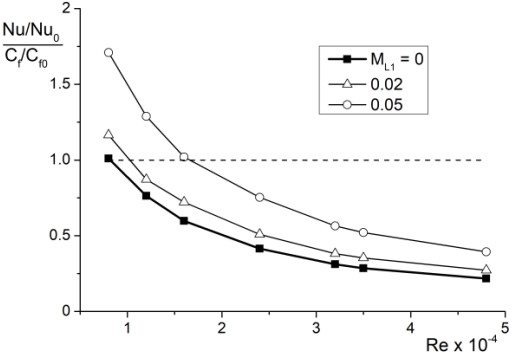

**Figure 11.** The effect on the thermal hydraulic performance parameter of the Reynolds number of the gas flow and the mass fraction of the dispersed phase at the inlet. $d_1$ = 15 μm.

## 5. Comparison with Results of Other Authors

Comparisons with the data of LES simulations of a solid particle-laden flow around a two-dimensional obstacle were made according to the conditions of [10]. The following data were used for the comparative analysis: $h = H/7$, $V_{P1} = U_{m1}/25$, the Reynolds number was plotted from the obstacle height, Re $= HU_{m1}/\nu = 740$, $\rho_P/\rho = 769.2$, and $\rho_P$ was the particle material density. The height of the boundary layer for a single-phase flow in the inlet section of the computational domain was $\delta = 7h$, and the carrier phase was atmospheric air at $T = 293$ K (see Figure 12). Here, $h = 7$ mm was the obstacle height, $H = 1$ mm, $U_{m1} = 1.59$ m/s was the free flow velocity, and $V_{P1} = 0.06$ m/s. The two-dimensional obstacle was square in cross section and was mounted on the bottom wall. The flow of solid particles was blown vertically through a flat slot along the normal surface at distance $h$ from the trailing edge of the obstacle. The number of solid particles during the LES calculation was $2 \times 10^5$. The calculations were performed for three Stokes numbers, St$^+ = \tau u_*^2/\nu = 0.25$, 1, 5, and 25, where $\tau = \rho_P d^2/(18\mu)$ was the particle relaxation time and $u_* = 0.5$ m/s was the friction velocity for a single-phase flow without particles, other things being equal. This corresponded to the solid particle diameters $d = 8$, 15, 34, and 76 μm. The calculations were carried out in a two-dimensional formulation for an isothermal two-phase flow around a single obstacle.

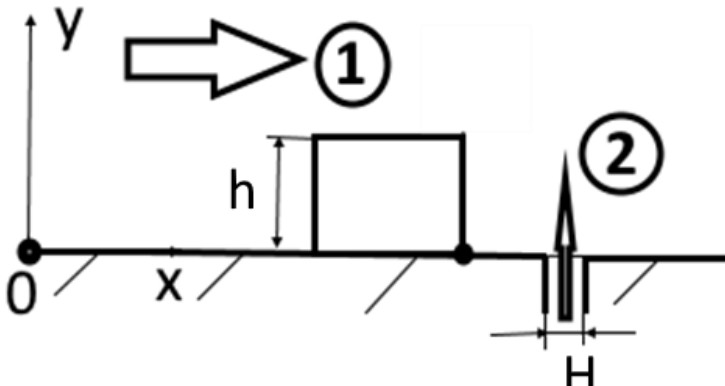

**Figure 12.** Scheme of two-phase flow behind a 2D square obstacle. The 1 is the single-phase air flow with the mean mass velocity $U_{m1}$, and the 2 is the dispersed phase stream $U_{P1}$.

The profiles of the dispersed phase concentration in the near-wall zone of the plate at $y = 0.02h$ are shown in Figure 13. Here, $C_b$ is the mean concentration of particles over the hole (slot) width at the inlet to the computational domain. As the Stokes number, St$_+$, increased, heavier particles stopped penetrating into the recirculation region, resulting in lower concentrations along the obstacle wall. The local maximum concentration at $x/h \approx 1$ for all studied Stokes numbers (particle diameters) is explained by the injection of the particle flow. A characteristic feature of the low-inertia particles was a significant increase in the concentration of particles near the obstacle wall, according to the LES data ($C/C_b \approx 10$–20). Most likely, such an accumulation of particles in the corner near the wall of the obstacle can be explained by the effect of the accumulation of particles in [42]. For our Eulerian simulations, an increase in concentration was also obtained, but the values were much smaller (by a factor of approximately 8–10). For inertial particles at St$_+ = 5$, the region turned out to be almost completely free of solid particles. This was typical for both the data of the LES calculations [10] and our numerical calculations. Behind the obstacle, a decrease in the particle concentration in the near-wall region was observed, and here our numerical predictions agreed satisfactorily with the LES data (the difference did not exceed 20% at St$_+ = 1$ and 5 and did not exceed 100% at St$_+ = 0.25$).

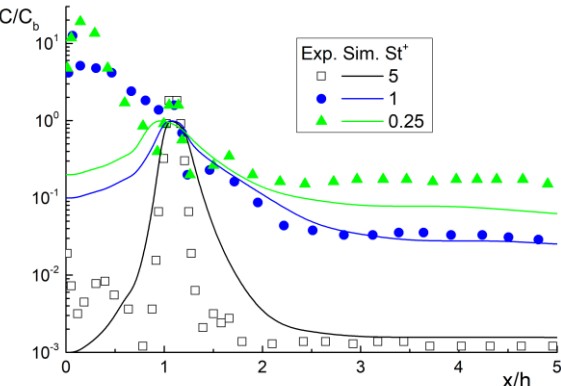

**Figure 13.** Concentration profiles of the dispersed phase at $y = 0.02h$ vs. the Stokes number $St_+$ along the length of the channel behind a 2D obstacle. The points are LES calculations [10]; the lines are the authors' predictions. $h = H/7$, $V_{P1} = U_{m1}/25$, Re $= HU_{m1}/\nu = 740$, $\rho_P/\rho = 769.2$. Reprinted with permission from (Grigoriadis, D.G.E. et al.)

Figure 13 shows the concentration profiles of the dispersed phase when the Stokes number, $St_+$, is varied along the length of the channel behind a two-dimensional obstacle. Particles at $St_+ = 0.25$ accumulated in the near-wall region near the bottom wall. Further downstream, heavier particles gradually left the recirculation region, and at $St_+ > 1$ the decrease in their distribution profile was similar to a Gaussian distribution. For the largest particles at $St_+ = 25$, according to the results of our numerical predictions, an underestimation of the position of the concentration maximum was observed, and in general the particles rose lower than according to the LES results [10].

Figure 14 shows the profiles of the dispersed phase concentration when the Stokes number, $St_+$, was varied along the length of the duct behind a two-dimensional obstacle. Particles at $St_+ = 0.25$ accumulated in the near-wall region near the bottom wall. Further downstream, heavier particles gradually left the recirculation region, and at $St_+ > 1$ the decrease in their distribution profile was similar to a Gaussian distribution. An underestimation of the position of the concentration maximum was observed, according to the results of our numerical predictions for the largest particles at $St_+ = 25$. The maximal penetration coordinate in the transverse directions in our RANS predictions was smaller than that in the LES results [10].

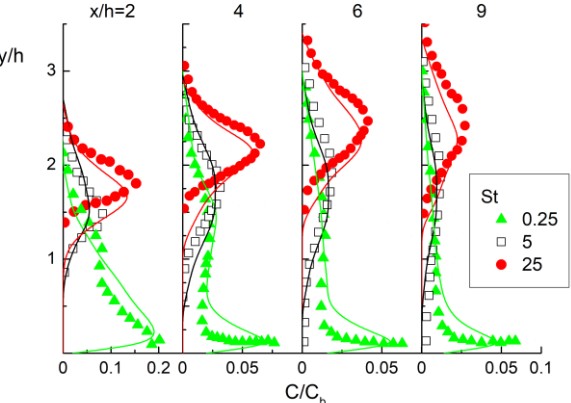

**Figure 14.** The transverse profiles of particle concentrations for various Stokes numbers, $St_+$, after a 2D obstacle. The points are LES calculations [10]; the lines are the authors' predictions.

## 6. Conclusions

Two-dimensional numerical simulations of the local flow structure, turbulence, and heat transfer in a ribbed flat duct during the evaporation of water droplets in a gas flow

were carried out. The set of steady-state RANS equations written with consideration of the influence of droplet evaporation on the transport processes in gas is used. The two-fluid Eulerian approach was used to describe the flow dynamics and heat and mass transfer in the dispersed phase. To describe the turbulence of the gas phase, an elliptical blending RSM model was employed.

It was shown that the transverse profiles of the averaged longitudinal velocity components of the gaseous and dispersed phases were similar to those of the single-phase flow regime. The gas velocity in the gas-droplet flow was slightly (≤3%) higher than those in the single-phase flow. The droplet velocity is higher than the gas-phase velocity in the two-phase flow. Finely dispersed droplets (Stk < 1) penetrated well into the region of flow recirculation and were observed over the entire cross section of the duct. They could leave the region between the two ribs due to their low inertia. Large inertial droplets (Stk > 1) were present only in the mixing layer and the flow core and accumulated in the near-wall region close to the downstream wall of the rib. A significant increase in heat transfer (up to 2.5 times) in comparison with a single-phase flow in a ribbed duct was shown when evaporating water drops were added to a single-phase gas turbulent flow. For the most inertial droplets, which were not involved in the separation motion in the region of the main recirculation zone behind the BFS (upstream rib), the heat transfer intensification decreased and trended toward the corresponding value for the single-phase flow regime in the recirculation zone. An increase in heat transfer was obtained behind the reattachment point. A decrease in heat transfer was shown in the zone close to the FFS (downstream rib).

**Author Contributions:** Conceptualization, M.A.P. and V.I.T.; methodology, M.A.P. and V.I.T.; investigation, M.A.P.; data curation, M.A.P. and V.I.T.; formal analysis, M.A.P. and V.I.T.; writing—original draft preparation, M.A.P. and V.I.T.; writing—review and editing, M.A.P. and V.I.T.; resources, M.A.P. and V.I.T.; project administration, V.I.T.; All authors have read and agreed to the published version of the manuscript.

**Funding:** This work was partially supported by the Ministry of Science and Higher Education of the Russian Federation (mega-grant 075-15-2021-575). The RANS and turbulence models for the single-phase turbulent flow were developed under a state contract with IT SB RAS (121031800217-8).

**Acknowledgments:** The authors thank Sebastian Ruck (Karlsruhe Institute of Technology, Karlsruhe, Germany) for providing the experimental database in an electronic form.

**Conflicts of Interest:** The authors declare no conflict of interest.

## Nomenclature

| | |
|---|---|
| $C_f/2 = \tau_W/\left(\rho U_{m1}^2\right)$ | wall friction coefficient |
| $D$ | droplet diameter |
| $H$ | rib height |
| $2k = \langle u_i' u_i' \rangle$ | turbulent kinetic energy |
| $M_L$ | mass fraction |
| $\text{Nu} = -(\partial T/\partial y)_W H/(T_W - T_m)$ | Nusselt number |
| $p$ | rib pitch |
| $q_W$ | heat flux density |
| $\text{Re}_D = U_m D_h/\nu$ | Reynolds number, based on hydraulic diameter |
| $\text{Re} = U_{m1}H/\nu$ | Reynolds number, based on the duct height |
| $\text{Stk} = \tau/\tau_f$ | the mean Stokes number |
| $T$ | temperature |
| $\mathbf{U}_L$ | the mean droplet velocity |
| $U_{m1}$ | mean mass flow velocity |
| $\mathbf{U}_S$ | the fluid (gas) velocity seen by the droplet |
| $u_*$ | wall friction velocity |
| $x$ | streamwise coordinate |
| $x_R$ | position of the flow reattachment point |
| $y$ | distance normal from the wall |

| Subscripts | |
|---|---|
| 0 | two-phase mist flow in a smooth duct |
| 1 | initial condition |
| $W$ | wall |
| $L$ | liquid |
| $M$ | mean mass |
| Greek | |
| $\Phi$ | volume fraction |
| $P$ | density |
| $\mu$ | the dynamic viscosity |
| $\nu$ | kinematic viscosity |
| $\tau$ | the droplet relaxation time |
| $\tau_W$ | wall shear stress |
| Acronym | |
| BFS | backward-facing step |
| CV | control volume |
| FFS | forward-facing step |
| THE | heat transfer enhancement |
| RANS | Reynolds-averaged Navier–Stokes |
| SMC | second-moment closure |
| TKE | turbulent kinetic energy |

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
