# Peer review of "RANS Modeling of Turbulent Flow and Heat Transfer in a Droplet-Laden Mist Flow through a Ribbed Duct"

_water, doi:10.3390/w14233829_

Round 1

Reviewer 1 Report

The paper investigates turbulent flow and heat transfer in a droplet-laden mist flow through a ribbed duct by using the classical RANS modeling. Euler approach is used to simulate flow, heat and mass transfers for the air and the dispersed phases while  evaporation of droplets that affects the two-phase processes is taken into account using the sink and source terms. The numerical model and method used are well validated against the other models under different problems. The RANS turbulence model and numerical method used have been well-established and extensively found in literature. Although the content of the paper is not very significant, it is of scientific interest and useful for an academic viewpoint. The paper may be publishable in the Water J. provided that the following suggestions are reasonably addressed.

- In the Mathematical model section, it is said that droplet volume fraction at the inlet is low according to preliminary estimations of the authors. Please clarify what is the preliminary estimations.

-Why does the proposed model not perform very well at low Stokes number and at small x/h (near the inlet)? How could the model/approach be improved? 

Author Response

Reviewer#1

The paper investigates turbulent flow and heat transfer in a droplet-laden mist flow through a ribbed duct by using the classical RANS modeling. Euler approach is used to simulate flow, heat and mass transfers for the air and the dispersed phases while evaporation of droplets that affects the two-phase processes is taken into account using the sink and source terms. The numerical model and method used are well validated against the other models under different problems. The RANS turbulence model and numerical method used have been well-established and extensively found in literature. Although the content of the paper is not very significant, it is of scientific interest and useful for an academic viewpoint. The paper may be publishable in the Water J. provided that the following suggestions are reasonably addressed.

We very much appreciate Reviewer#1 for the encouragement and thought-provoking comments, which are truly valuable for us to improve the quality of this paper. Having considered the review we would like to note the following. We thank the Reviewer for thorough work, which has improved the content of our paper. Our responses to the reviewer are presented below and colored by yellow in the paper.

- In the Mathematical model section, it is said that droplet volume fraction at the inlet is low according to preliminary estimations of the authors. Please clarify what is the preliminary estimations.

Response. Thank you for the comment.

We did not say that the droplet volume fraction at the inlet is low according to our preliminary evaluations. We meant that the droplet volume fraction at the inlet is low (Φ1 = ML1r/rL < 2´10-4) and because we did not take into account the droplets’ break-up and coalescence.

We carried out the preliminary simulations with and without the droplets break-up and coalescence. The difference was up to 2%, but the computational time was increased up to 50%.

For the sake of clarity, we have removed this part of the sentence.

 Before correction

The effect of break-up and coalescence in the two-phase mist flow can be neglected due to a low droplet volume fraction at the inlet Φ1 = ML1r/rL < 2´10-4 according to preliminary estimations of the authors.

After correction

The effect of break-up and coalescence in the two-phase mist flow can be neglected due to a low droplet volume fraction at the inlet Φ1 = ML1r/rL < 2´10-4.

-Why does the proposed model not perform very well at low Stokes number and at small x/h (near the inlet)? How could the model/approach be improved? 

Response. It is difficult to give an unambiguous answer to this question. The reasons may lie both in the inaccuracies of our RANS mathematical model and inaccuracies of LES simulations [10]. The method of particles injection in LES [10] has been markedly differed from our main numerical predictions.

The profiles of the mean longitudinal velocity component in the single-phase flow agree well with the experimental data (the difference does not exceed 5–7%). The agreement between measurements and numerical predictions for longitudinal velocity pulsations is also quite good (the difference does not exceed 10%) except for the near-wall region.

The heat transfer distributions in the single-phase flow agree satisfactorily over most of the inter-rib cavity (the difference does not exceed 15%). The exception is the section at x/h < 2.

It seems that the possible ways to improve the models are performing of the DES or LES simulations for the two-phase mist flow. These data can give the best comparison with the experimental results of this complicated case. LES approach with taking into account the effect of the dispersed phase on sub-grid scale stresses requires high-performance supercomputers and these predictions have high computational cost and time. It limits the application of this method for engineering computations. RANS still remains a robust tool for engineering simulations and this method can significantly reduce computational costs and time and makes parametric studies feasible.

 Authors,

Maksim Pakhomov, Viktor Terekhov.

Reviewer 2 Report

The work is of high quality and can be accepted for publication after some modifications.

1. I suggest the authors include governing equations in the manuscript. The evaporation model is not found in Section 2.

2. More visualization of fluid and temperature fields should be added to make readers easier to follow.

3. Can the authors explain the origination of difference between experiment and their numerical results?

4. High-resolution figures are expected. Current versions are not clear enough.

5. 

Author Response

Reviewer#1

The work is of high quality and can be accepted for publication after some modifications.

 We very much appreciate Reviewer#2 for the encouragement and thought-provoking comments, which are truly valuable for us to improve the quality of this paper. Having considered the review we would like to note the following. We thank the Reviewer for thorough work, which has improved the content of our paper. Our responses to the reviewer are presented below and colored by yellow in the paper. We have followed the review’s suggestions on the descriptions of the model. To make this paper more readable and self-contained, Eqs. (1)-(10) were added in the revised manuscript.

  1. I suggest the authors include governing equations in the manuscript. The evaporation model is not found in Section 2.

Response. We agree with the Reviewer remark.

We added the Section with governing equations for the both gas and dispersed phases, Reynolds stress turbulence model and evaporation model (pages 3-5).

  1. More visualization of fluid and temperature fields should be added to make readers easier to follow.

Response. Thanks for the reviewer’s valuable comment We added the plots of streamwise velocity and temperature of the gas phase to the paper (see Fig. 9).

  1. Can the authors explain the origination of difference between experiment and their numerical results?

Response. Thank you for the comment. We added some explanation on the page 8.

It is difficult to give an unambiguous answer to this question. The reasons may lie both in the inaccuracies of our mathematical model and inaccuracies of PIV measurements, especially in the near-wall zone of the channel.

The profiles of the mean longitudinal velocity component in the single-phase flow agree well with the experimental data (the difference does not exceed 5–7%). The agreement between measurements and numerical predictions for longitudinal velocity pulsations is also quite good (the difference does not exceed 10%) except for the near-wall region.

The heat transfer distributions in the single-phase flow agree satisfactorily over most of the distance between two obstacles (the difference does not exceed 15%). The exception is the section at x/h < 2.

  1. High-resolution figures are expected. Current versions are not clear enough.

Response. We tried to improve the quality of figures and hope the quality is increased of figures. Perhaps it was after formatting from .docx format to pdf format.  

 Authors,

Maksim Pakhomov, Viktor Terekhov.

Reviewer 3 Report

see attached report

Author Response

Reviewer#3

We very much appreciate Reviewer#3 for the encouragement and thought-provoking comments, which are truly valuable for us to improve the quality of this paper. Having considered the review we would like to note the following. We thank the Reviewer for thorough work, which has improved the content of our paper. Our responses to the reviewer are presented below and colored by yellow in the paper.

  1. The authors do not present the basic equations that numerically solved. I think this is a must as only by that a physically motivated discussion of results is possible. For instance, information about the modeling of evaporation is missing. Thermodynamically, phase changes are governed by the Clapeyron equation that expresses the fact that Gibbs energy remains constant. However, proper inspection shows that evaporation strongly depends on both local pressure and temperature.

It is not clear at all how these effects are treated in both the averaged equations and the turbulent transport equation.

Response. Thanks for the reviewer’s valuable comment.

We added the Section with governing equations for the both gas and dispersed phases, Reynolds stress turbulence model and evaporation model (pages 3-5).

About the evaporation model. We used the simple evaporation model of B. Abramzon and W.A. Sirignano (Int. J. Heat Mass Transfer, 1989). This model did not use the term of Gibbs energy. The model of Abramzon and Sirignano was based on the so called ‘film’ theory. The film theory assumes that the resistance to heat or mass transfer between a droplet surface and a gas flow may be modelled by introducing the concept of gas films of constant thicknesses The key concepts of this theory are film.

  1. There is no justification given why the simulations are restricted to 2D. It is well known that in certain parameter regimes the flow may get unstable against perturbation in the spanwise direction.

Response. We have performed the simulation for a 2D two-phase mist case. Keshmiri A. (Heat Mass Transfer, 2012) was found that flow and heat transfer in a single-phase flow over the centerline of the 3D ribbed channel studied by Rau et al. (ASME J. Turbomach., 1998) can be represented by a 2D configuration with relatively good accuracy.

 Keshmiri A. Numerical sensitivity analysis of 3-and 2-dimensional rib-roughened channels. Heat Mass Transfer, 2012. V. 48(7) 15 pages. DOI: 10.1007/s00231-012-0968-z

Rau G., Cakan M., Moeller D., Arts T. The effect of periodic ribs on the local aerodynamic and heat transfer performance of a straight cooling channel. ASME J. Turbomach. 1998. V. 120. P. 368–375. https://doi.org/10.1115/1.2841415

 In the recent experimental work of Huang et al. (ASME J. Thermal Sci. Eng. Appl., 2017) was shown that the distribution of the Nusselt number in the transverse direction for the two-phase mist flow in a ribbed duct has a uniform character, and only at the side walls does a decrease in heat transfer occur.

Huang Y.-H., Chen C.-H., Liu Y.-H. Nonboiling heat transfer and friction of air/water mist flow in a square duct with orthogonal ribs. ASME J. Thermal Sci. Eng. Appl. 2017. V. 9. Paper 041014.

 Authors of these works for single-phase and two-phase mist flows did not find the flow instability by appearance of perturbation in the spanwise direction.

  1. It is not stated how effects of gravity are treated, Usually, in forced convection such effects are neglected but authors claim that gravity may explain some deviations of results. So also here more information is needed.

Response. The effect of gravity (droplet inertia) will affect only the dynamics of the dispersed phase. Of course, the effect of gravity on motion and heat transfer (forced convection) in the gas phase is, negligible. We did not take it into account.

We have slightly modified the sentence on the page 10.

  1. Moreover, the assumption that droplets are not allowed touch walls need justification.

Response. We thank the reviewer for this comment.

For the dispersed phase on the duct wall, the boundary condition of the “absorbing wall” [30] was used, when a droplet after contact with the wall surface does not return to the flow. All droplets deposited from two-phase flow onto the wall are momentarily evaporate. Thus, the pipe surface is always dry and there is no liquid film or spots of deposited droplets formed on the wall [25,31,35]. This assumption for the heated surface is valid (see, for example, papers [25,35]). Furthermore, this condition is valid if the temperature difference between the wall and the droplet is greater than[37].

 A proper discussion of the dimensionless parameters is missing. What is about Jakob number that characterizes phase change processes?

Response. Thanks for the reviewer’s valuable comment

Yes, we did not use in this model the dimensionless equations and parameters. We understand the importance of employing non-dimensional parameters. It is convenient for the numerical analysis using the dimensional parameters at the inlet of the duct.   

The Jacob number Ja = CP(T–TL)/L (it is the ratio of sensible heat to latent heat during droplets evaporation) characterizes the intensity of evaporation processes and is the reciprocal of the Kutateladze number Ku. In our cases for our conditions Ja number is Ja ≤ 0.01. We have not explicitly used this important criterion, but it is implicitly inserted into the droplet evaporation model (see page 4).

 Weber number is incorrectly defined.

Response. Thank you for the carefully reading of the manuscript. The typo was corrected.

  1. Blow-ups of the flow in the corner region would be helpful as well as information about vorticity,

Response. We added two stations (x/h = 0.5 and 8.5) in the Figs. 7a-7d. These cross-sections are located close to the upwind and downstream ribs.

We added the Fig. 7d with the transverse profiles of vorticity along the inter-rib cavity and on the top-wall of the downstream rib. Some explanations were added on the page .

In conclusion: The scientific content of the manuscript is of some interest. Assumptions made must be justified on a physical basis. Full equations and parameters should be shown and discussed. Therefore, I recommend revising the manuscript.

 Authors,

Maksim Pakhomov, Viktor Terekhov.

Round 2

Reviewer 2 Report

The authors have addressed my concerns and the manuscript can be accepted in present form.

The authors' efforts are appreciated.

Author Response

Dear Reviewer thank you for positive opinion about our paper.

Reviewer 3 Report

The authors resubmitted the manuscript after conducting some revision suggested by the referee. I still disagree in the point of restriction to a 2D modelling. Any turbulent flow is inherentely of 3D nature. When bounded by side walls the 3D charcter of flow is obivous. But also in a gemoetrically dtirct 2D arrabgement, flow and boundary layers may develop 3D charactweristics to to exitation of secondary Tollimien Schlichting waves, especially when wall geometry varies. Therefore, the simple statement of the authors that they consider 2D because flow is 2D is strongly questionable. I would recommend publication of the manzuscript when a clear sttement as "In this paper the numerical simulations are restricted to 2D" would be given. Of course, RANS methods are not well suited to investiagte 3D effects. Here, a more subtile modelleing according to LES must be applied. To summarize: It is up to the Editor to decide if this cruicial point is addressed in a physicallly satisfying way by authors. In my opinion it is not.

Author Response

Reviewer#3

Having considered the review we would like to note the following. Our responses to the reviewer are presented below..

I still disagree in the point of restriction to a 2D modelling. Any turbulent flow is inherentely of 3D nature. When bounded by side walls the 3D charcter of flow is obivous. But also in a gemoetrically dtirct 2D arrabgement, flow and boundary layers may develop 3D charactweristics to to exitation of secondary Tollimien Schlichting waves, especially when wall geometry varies. Therefore, the simple statement of the authors that they consider 2D because flow is 2D is strongly questionable. I would recommend publication of the manzuscript when a clear sttement as "In this paper the numerical simulations are restricted to 2D" would be given. Of course, RANS methods are not well suited to investiagte 3D effects. Here, a more subtile modelleing according to LES must be applied. To summarize: It is up to the Editor to decide if this cruicial point is addressed in a physicallly satisfying way by authors. In my opinion it is not.

We absolutely agree with the reviewer that turbulent flow is 3D in nature. In this case, there are many cases when it is possible to use the 2D approach to describe a quasi-two-dimensional flow. For example, if the channel width Z is much greater than its height H (Z/H > 10). For example, in a well-known experimental work on the study of turbulent solid particle-laden flow in a backward-facing step in a flat channel.

Fessler and Eaton (JFM, 1999): “The aspect ratio of the sudden expansion (width/step height) was 17:1, which was sufficient to ensure essentially two-dimensional flow throughout a significant portion of the test section” (page 99).

 It is obvious that near the side walls this approach is wrong and leads to large errors, but for most of the duct cross-section and especially for the axial zone of the duct for a single-phase flow, it is quite applicable. This is also discussed in Keshmiri (Heat Mass Transfer, 2012) (see Figures 9 and 13).

Keshmiri A. (Heat Mass Transfer, 2012) was found:

“…that the present results for a 3D channel are in relatively good agreement with the data. It was also shown that a 2D channel can be used to represent the flow in the centre-line of a 3D channel with relatively good accuracy.” (page 1 of the paper)

 I have to repeat the text from our first answer to Reviewer#3:

Flow and heat transfer in a single-phase flow over the centerline of the 3D ribbed channel studied by Rau et al. (ASME J. Turbomach., 1998) can be represented by a 2D configuration with relatively good accuracy.

 Fessler J.R., Eaton J.K. Turbulence modification by particles in a backward-facing step flow. J. Fluid Mech. 1999. V. 314. P. 97-117. DOI: 10.1017/S0022112099005741

Keshmiri A. Numerical sensitivity analysis of 3-and 2-dimensional rib-roughened channels. Heat Mass Transfer, 2012. V. 48(7) 15 pages. DOI: 10.1007/s00231-012-0968-z

Rau G., Cakan M., Moeller D., Arts T. The effect of periodic ribs on the local aerodynamic and heat transfer performance of a straight cooling channel. ASME J. Turbomach. 1998. V. 120. P. 368–375. DOI: 10.1115/1.2841415

 We performed the numerical simulation based on using our in-house 3D Eulerian code (see the attached file). All predictions were carried out on a “medium” grid containing 256´120´120 = 3 686 400 control volumes (CVs). Additionally, simulations were carried out on grids containing “coarse” 128´60´60 = 460800 and “fine” 400´150´150 = 9 Mio CVs. The first computational cell was located at a distance from the wall y+ = u*y/ν » 0.5 (the friction velocity u* was determined for a single-phase air flow with other identical parameters).

All simulations were performed for the flow around the system of the 2nd and 3rd obstacles. The computational domain included two square ribs with height h = 4 mm. The height of a smooth square duct was H = Z = 40 mm (H/h = Z/h = 10), and the distance between two ribs was p/h = 5–12, where Z is the duct width. The mass-average gas velocity in the inlet cross-section in the computational domain varied within Um1 = 5 m/s, and the Reynolds number for the gas phase, constructed from the mass-average gas velocity at the inlet and the duct height, was ReH = HUm1/n » 1.33´104. The initial average droplet diameter was d1 =5–50 µm and their mass fraction was ML1 = 0–5%. The initial temperature of the gaseous and dispersed phases was T1 = TL1 = 293 K.

The distributions of heat transfer along the duct width are given in the Figure. Here station z/d = 0 and 5 are the duct centerline and side wall coordinate respectively. The difference between the results obtained using 2D and 3D model for the two-phase mist flow is up to 5% for the most part of the duct. These data did not include to the manuscript due to the unambiguity of understanding main aims of the paper.

Figure. The effect of 2D (dashed lines) and 3D (solid lines) models on distributions of heat transfer along the duct width in the single-phase and droplet-laden mist flows in the ribbed duct.

 About the effect of the secondary Tollmien-Schlichting waves on the numerical results and flow instability. It is complicated question. In all abovementioned papers this task did not discuss and studied experimentally or numerically. We used in our predictions the steady-state approach and we cannot get this phenomenon. 

 Authors,

Maksim Pakhomov, Viktor Terekhov.
